# Spindle Position Checkpoint Kinase Kin4 Regulates Organelle Transport in *Saccharomyces cerevisiae*

**DOI:** 10.3390/biom13071098

**Published:** 2023-07-10

**Authors:** Lakhan Ekal, Abdulaziz M. S. Alqahtani, Maya Schuldiner, Einat Zalckvar, Ewald H. Hettema, Kathryn R. Ayscough

**Affiliations:** 1School of Biosciences, University of Sheffield, Sheffield S10 2TN, UK; lakhan.ekal@embl-hamburg.de (L.E.); amsalqahtani1@sheffield.ac.uk (A.M.S.A.); e.hettema@sheffield.ac.uk (E.H.H.); 2Department of Biology, Faculty of Science, University of Bisha, P.O. Box 551, Bisha 61922, Saudi Arabia; 3Department of Molecular Genetics, Weizmann Institute of Science, Rehovot 7610001, Israel; maya.schuldiner@weizmann.ac.il (M.S.); einat.zalckvar@weizmann.ac.il (E.Z.)

**Keywords:** organelle transport, spindle position checkpoint (SPOC), mitotic exit network (MEN), peroxisome, actin cytoskeleton, class V myosin

## Abstract

Membrane-bound organelles play important, frequently essential, roles in cellular metabolism in eukaryotes. Hence, cells have evolved molecular mechanisms to closely monitor organelle dynamics and maintenance. The actin cytoskeleton plays a vital role in organelle transport and positioning across all eukaryotes. Studies in the budding yeast *Saccharomyces cerevisiae* (*S*. *cerevisiae*) revealed that a block in actomyosin-dependent transport affects organelle inheritance to daughter cells. Indeed, class V Myosins, Myo2, and Myo4, and many of their organelle receptors, have been identified as key factors in organelle inheritance. However, the spatiotemporal regulation of yeast organelle transport remains poorly understood. Using peroxisome inheritance as a proxy to study actomyosin-based organelle transport, we performed an automated genome-wide genetic screen in *S. cerevisiae*. We report that the spindle position checkpoint (SPOC) kinase Kin4 and, to a lesser extent, its paralog Frk1, regulates peroxisome transport, independent of their role in the SPOC. We show that Kin4 requires its kinase activity to function and that both Kin4 and Frk1 protect Inp2, the peroxisomal Myo2 receptor, from degradation in mother cells. In addition, vacuole inheritance is also affected in *kin4*/*frk1*-deficient cells, suggesting a common regulatory mechanism for actin-based transport for these two organelles in yeast. More broadly our findings have implications for understanding actomyosin-based transport in cells.

## 1. Introduction

Eukaryotic cells contain a set of membrane-delimited compartments, called organelles. A full complement of organelles is important for cellular functions and viability. The organelles containing genetic material, such as mitochondria, chloroplast, and the nucleus, must be inherited as they cannot form de novo. Although some organelles may form de novo, such as peroxisomes, lysosomes, and lipid droplets, they are also actively partitioned. This ensures rapid cellular activation of all organelle-associated functions in daughter cells and more competitive growth or cell survival under various conditions. Hence, molecular mechanisms have evolved to secure proper organelle partitioning between daughter cells during cell division [1,2]. Besides inheritance, organelles are transported either along microtubules or actin cables to site functions. Disruption of organelle transport in humans can lead to various disorders, including Griscelli syndrome, which is caused by a defect in actomyosin-based melanosome transport [3].

Asymmetric cell division requires ordered partitioning of organelles between mother and daughter cells. This process is extensively studied in the yeast *Saccharomyces cerevisiae* (*S*. *cerevisiae*; from here on, also simply termed yeast), which grows asymmetrically by forming a bud. Proper partitioning of organelles relies on a balance of retention in the mother and transport of organelles to the emerging bud. Molecular dissection in *S*. *cerevisiae* has uncovered common principles of organelle positioning and partitioning [1,2,4]. The actin cytoskeleton plays a central role to set up and maintain cell polarity, and the polarised actin cables direct organelle transport from the mother cell to the bud [2]. Most yeast organelles, including mitochondria, vacuoles (the yeast lysosome), peroxisomes, and the endoplasmic reticulum (ER), recruit an unconventional class V myosin Myo2 or Myo4 via organelle-specific receptors within the mother cell in the early G1 phase [2] (Appendix A). Transport along actin cables is initiated upon bud emergence and continues throughout G1 [4]. Upon arrival in the bud, organelles release myosin. Positioning in the daughter is then achieved by association with other structures [2,5].

Disassembly of Myo2 from organelles has been studied in detail for vacuoles. The Myo2 receptor, Vac17, is phosphorylated in the mother cell and recruits the E3 ubiquitin ligase Dma1 or its paralogue Dma2. However, Dma1/2 activity is suggested to be stimulated only upon entry of the vacuole into the bud of large-budded cells [6]. This is, in part, controlled by the evolutionarily conserved p21-activated kinases (PAKs) Cla4 and Ste20. The activity of these two paralogue kinases is restricted to the bud [7]. However, upon Cla4 overexpression, vacuole inheritance is impaired as potentially Dma1/2-dependent Vac17 breakdown is initiated prematurely [8]. Overexpression of Cla4 also affects mitochondrial inheritance, and Dma1/2/Cla4-dependent breakdown of the mitochondrial Myo2 receptor Mmr1 has also been described [9] (Appendix A).

Myo2 is also required for the initial positioning of the nucleus in the bud neck [10], but subsequent movement into the bud requires a microtubule-based system [11,12]. The correct positioning of the nucleus is controlled by the spindle position checkpoint (SPOC), which blocks entry to mitotic exit when spindle pole bodies (SPBs) fail to enter the bud during anaphase [11,13,14,15]. The ras-like GTPase Tem1, in a GTP-bound form, triggers a kinase cascade that leads to mitotic exit through the activation of the phosphatase Cdc14 [16,17].

Key proteins of the SPOC are the Tem1′s GTPase-activating protein (GAP) complex Bfa1-Bub2 and the Kin4 kinase [18] (Appendix A). These proteins determine whether the mitotic exit network (MEN) is activated by regulating the nucleotide-bound state of Tem1 [19] and its localisation at SPBs [20]. Bfa1-Bub2 GAP activity is tightly regulated. In the mother cell, Kin4 phosphorylates Bfa1 and activates the Bfa1-Bub2 complex by preventing inactivation through Bfa1 phosphorylation by the Polo kinase Cdc5 [21,22]. This results in the inactivation of Tem1 in the mother cell. Kin4 kinase activity requires its phosphorylation by the Elm1 kinase [23,24]. In the daughter cell, Lte1 inactivates Kin4, permitting the inactivation of Bfa1-Bub2 by Cdc5 and subsequent activation of the MEN during anaphase [25,26]. Lte1 is also proposed to directly activate Tem1 [27]. This spatial setup of a negative regulator of the MEN in the mother cell (Kin4) and an activator of the MEN in the daughter cell (Lte1) is used to monitor the position of the two SPBs. Besides spatial control of MEN activation, the timing of the MEN is also tightly regulated [28,29]. Cla4 and Ste20 are required for MEN activation and Cla4 acts upstream of Lte1 [25,30] (Appendix A).

In dividing yeast cells, peroxisomes multiply by growth and division before they segregate with high fidelity between mother and daughter cells. The balance between transport to the bud and retention in the mother cell determines proper segregation [31]. Two peroxisomal proteins, Inp1 and Inp2, are specifically required for the inheritance of peroxisomes. Inp1 binds peroxisomes via Pex3 [32,33] and mediates retention in the mother cell [34] by attaching peroxisomes to the plasma membrane [35,36]. Inp2 acts as the peroxisomal Myo2 receptor [37]. Pex19 in complex with Inp2 contributes to peroxisome delivery to the bud [38,39]. Both Pex3 and Pex19 are also required for peroxisome biogenesis [40]. The protein levels of Inp2 are regulated as the cell cycle progresses. It reaches the peak around 100 min after resumption from the G1 phase [37]. Myo2 cargo-binding domain mutations, which disrupt the interaction with Inp2, affect peroxisome transport to the bud and result in an increase in Inp2 levels [41]. Dma1/2 affects peroxisome positioning in the bud analogous to vacuole positioning [6]. This suggests that peroxisome and vacuole inheritance are controlled through similar regulatory mechanisms.

Peroxisome fission is mediated by the dynamin-related proteins (DRPs) Vps1 and Dnm1. In *vps1Δdnm1Δ* cells, peroxisome fission is abolished and therefore in these cells, a single peroxisome is observed [42,43]. This peroxisome is anchored in the mother cell by Inp1 and pulled into the bud through the action of Inp2 [33]. However, even in *vps1Δdnm1Δ* cells, peroxisomes mainly multiply by fission. As the elongated peroxisome is positioned in the bud neck in most cells, it has been proposed that cytokinesis drives the fission of this single organelle [31,42]. We found that in a low percentage of *vps1Δdnm1Δ* cells, peroxisome segregation is affected and, in line with previous studies, in cells failing to inherit peroxisome, new peroxisomes form [43,44,45]. We hypothesised that using a starting strain with a challenging inheritance of peroxisomes in a genome-wide genetic screen may allow us to identify factors that have been missed in previous genetic screens concerning peroxisome biology. Through this approach, we identified Kin4 and its paralogue Frk1 as regulators of peroxisome and vacuole inheritance, acting independently of Bfa1 and therefore the SPOC. Our analyses indicate that Kin4 regulates peroxisome transport by protecting the peroxisomal Myo2 receptor Inp2 from premature degradation in the mother cell.

## 2. Materials and Methods

### 2.1. Yeast Strains and Plasmids

Yeast strains used in this study are listed in Appendix A, and they were derivatives of either BY4741 (*MAT**a** his3Δ1 leu2Δ0 met15Δ0 ura3Δ0*) or BY4742 (*MATα his3Δ1 leu2Δ0 lys2Δ0 ura3Δ0*) obtained from the EUROSCARF *S. cerevisiae* (S288C) strain collection. BY4741 and BY4742 were referred to as wild-type (WT) cells and used as isogenic controls throughout the paper. Double or triple gene deletions were generated by replacing the entire open reading frame (ORF) of the desired gene as described in [46,47]. The pFA6a-3 × HA-spHIS5 plasmid was used as a template for PCR to tag the *MYO2* open reading frame at the C-terminal in the genome with 3 × HA [48]. Triple mutants selected from the high-content screen that were used for further experimentation were checked for the presence of the correct gene deletions by PCR. Plasmids used in this study are listed in Appendix A. *URA3* and *LEU2* centromeric plasmids were derived from Ycplac33 and Ycplac111 [49] and contained the *PGK1* terminator. These *ARS1*/*CEN4* plasmids are present at 1–2 copies per cell [50].

Yeast expression plasmids were constructed either by gap repair mechanisms in yeast [51] or by conventional restriction digestion-ligation-based methods in *E. coli* [52]. Constitutive expression of the peroxisomal markers, HcRed-PTS1, mNG-PTS1, mKate2-PTS1, and mRuby2-PTS1, was under either the *HIS3* or *TPI1* promoter and the conditional expression of mNG-PTS1, Kin4, and Frk1 was under the *GAL1* promoter. Expression of Kin4 and Kin4-T209A was achieved through the *KIN4* promoter, and Inp2-ProtA was under the control of its own promoter. The Inp2ΔMIS mutant was generated by the deletion of 534–539 aa in *INP2* ORF.

### 2.2. Growth Conditions

Yeast cells were grown overnight, and on the next morning, they were diluted to an OD_600_ of 0.1 and were further grown to an OD_600_ of 0.5–0.6. Where the induction of a reporter protein was required, cells were transferred to a selective galactose medium at an OD_600_ of 0.3 and grown for the time indicated in the figures and text. Yeast cells were grown at 30 °C in one of the following mediums: rich YPD medium (1% yeast extract, 2% peptone, 2% glucose), yeast minimal medium 2 (YM2) for the selection of the uracil prototrophic marker (carbon source, 0.17% yeast nitrogen base without amino acids and ammonium sulphate, 0.5% ammonium sulphate, 1% casamino acids), or yeast minimal medium 1 (YM1) for the selection of all prototrophic markers (carbon source, 0.17% yeast nitrogen base without amino acids and ammonium sulphate, 0.5% ammonium sulphate). As carbon sources, 2% (*w*/*v*) glucose and galactose were added. The appropriate amino acid stocks were added to the minimal medium as required. In all, 10 OD_600_ units were collected at the selected time points as indicated in the figures and text. Cells were either analysed by immunoblotting or by fluorescence microscopy. For peroxisome quantification, the budding cells were considered single cells. The de novo peroxisome assays were performed as described previously [43].

### 2.3. Setup for Construction of Mutant Collecion and Microscopy Analysis

Synthetic genetic array (SGA)-based [53] methodology was employed to generate a library of haploid strains containing mutations in three genes (*vps1Δdnm1ΔxxxΔ*) and expressing the fluorescent protein mNeonGreen containing a peroxisomal targeting signal type 1 (mNG-PTS1), along with cytosolic mCherry. The mutant library generation and microscopy analysis were conducted as described in [54,55], respectively. Briefly, an SGA compatible strain, *vps1Δdnm1Δ* (mating type α), expressing mNG-PTS1 and cytosolic mCherry was constructed (see Appendix A). This strain was crossed on YPD rich medium plates with two libraries (both are mating type **a**) containing 5853 unique strains: (i) the library of single gene deletion mutants [56] and (ii) the Decreased Abundance by mRNA Perturbation (DAmP) library, containing hypomorphic alleles of essential genes [57]. The diploids were selected on a synthetic dropout (SD) medium containing antibiotics appropriate to both mating type α and A cells. The selected diploid cells were grown 5 days on nitrogen-starvation medium plates to induce sporulation. Spores with mating type α were selected on SD medium plates supplemented with canavanine (50 mg/L, Sigma-Aldrich, St. Louis, MO, USA) and thialysine (50 mg/L, Sigma-Aldrich, St. Louis, MO, USA),to select against remaining diploids. Finally, two rounds of selection were performed on SD medium plates containing canavanine, thialysine, and appropriate antibiotics to obtain haploid cells (mating type α) containing mutations in three genes (*vps1Δdnm1Δxxx∆*) and expressing mNG-PTS1, along with cytosolic mCherry. These mutants were grown in an SD medium overnight and then for 5 h during the day in a fresh medium, before visualising them using an Olympus microscope with a 60 × 0.9 NA Air lens and images from three different fields of view for each mutant strain were captured with a camera (ORCA-ER, Hamamatsu, Hamamatsu City, Japan). The microscopy images (~18,000) were analysed manually using Fiji-ImageJ-win64 software v1.53c [58]. Out of the 5853 mutant strains tested, 5057 strains achieved our quality cut-off of more than 60 cells within 3 microscopy images, and hence only these strains were considered for further analysis.

### 2.4. Image Acquisition

Cells were analysed with a microscope (Axiovert 200 M; Carl Zeiss, Oberkochen, Germany) equipped with an Exfo X-cite 120 excitation light source, band pass filters (Carl Zeiss and Chroma Technology, Rockingham, VT, USA), an α Plan-Fluar 100 × 1.45 NA and Plan-Apochromat 63 × 1.4 NA objective lens (Carl Zeiss), and a digital camera (Orca ER; Hamamatsu Photonics, Hamamatsu City, Japan). Image acquisition was performed using Volocity software version 7.0.0 (PerkinElmer, Waltham, MA, USA). Fluorescence images were collected as 0.5 μm z-stacks, merged into one plane using Openlab software version 5.5.2 (PerkinElmer), and processed further in Photoshop (Adobe, version 24.6). Bright-field images were collected in one plane and processed where necessary to highlight the circumference of the cells in blue. Each imaging experiment was performed at least three times, and representative images are shown. For quantitation analysis, 1–3 experiments were used. The graphs for quantitation analysis were plotted using GraphPad Prism 9 (GraphPad software, San Diego, CA, USA. www.graphpad.com accessed on 23 December 2022).

### 2.5. Time-Lapse Imaging

For time-lapse imaging, the samples were prepared into a glass bottom 35 mm μ-dish (Ibidi) containing an agarose gel pad, which was prepared by dissolving 2% (*w*/*v*) agarose (Geneflow, Lichfield, UK. A4-0700) in yeast minimal media. The cells were grown logarithmically and a 20 μL cell culture was immobilised under the agarose gel pad and spread uniformly by gently pressing the gel pad from the top. Fluorescence images were collected as 0.5 μm z-stacks for every 10 min time point. The images were processed to generate videos using Fiji-ImageJ-win64 software v1.53c [58].

### 2.6. Vacuolar Staining with FM4-64

To analyse vacuole inheritance, the vacuolar membrane was stained with FM4-64 (Invitrogen, Thermo-Fisher Scientific, Cambridge, UK. T3166) [59]. FM4-64 staining was performed as described previously [60], and the cells were imaged using an epifluorescence microscope.

### 2.7. Immunoblotting

For the preparation of yeast cell extracts by alkaline lysis, 10 OD_600_ equivalent cell cultures were centrifuged, and cell pellets were resuspended in lysis buffer (0.2 M NaOH and 0.2% β-mercaptoethanol) and left on ice for 10 min. The soluble protein was precipitated by the addition of 5% trichloroacetic acid (TCA) and incubation on ice for a further 15 min. Following centrifugation (13,000× *g*, 5 min, 4 °C), the pellet was resuspended in 10 μL 1 M Tris-HCl (pH 9.4) and 90 μL 1× SDS–PAGE sample loading buffer and boiled for 10 min at 95 °C. Samples (0.5–1OD_600_ equivalent) were resolved by SDS–PAGE, followed by immunoblotting. Blots were blocked in 2% (*w*/*v*) fat-free Marvel milk in TBS (50 mM Tris-HCl pH 7.6, 150 mM NaCl) containing Tween-20 (0.1%*v*/*v*). GFP-tagged proteins were detected using a monoclonal anti-GFP antibody (mouse IgG monoclonal antibody clone 7.1 and 13.1; 1:3000; Roche, Basel, Switzerland, 11814460001). Inp2–ProtA was detected by the peroxidase-anti peroxidase (PAP) antibody (rabbit; 1:4000; Sigma-Aldrich, Gillingham, Dorset, UK P1291). Vps1 was detected with the polyclonal anti-Vps1 antibody (rat; 1:10,000) [60]. Pgk1 was detected by a monoclonal anti-Pgk1 antibody (mouse; 1:7000; Invitrogen, Thermo-Fisher Scientific, Cambridge, UK 459250). Myo2-HA was detected by a monoclonal anti-HA antibody (mouse; 1:5000; Sigma-Aldrich, H9658). The secondary antibody was an HRP-linked anti-mouse polyclonal (goat; 1:4000; Bio-Rad, Watford, UK. 1706516) or HRP-linked anti-rat polyclonal (rabbit; 1:10,000, Sigma-Aldrich, A5795) antibody. Detection was achieved using enhanced chemiluminescence reagents (GE Healthcare, Hatfield, UK) and chemiluminescence imaging.

## 3. Results

### 3.1. vps1Δdnm1Δ Cells Show a Weak Peroxisome Segregation

DRP Vps1 and, to a lesser extent, Dnm1 mediate peroxisome fission in *S*. *cerevisiae*. Hence, in *vps1Δdnm1Δ* cells, one enlarged and elongated peroxisome is observed (Appendix A), frequently anchored on the mother side of the bud neck and pulled into the bud by Inp2/Myo2 [31,33,42]. Around the time of cytokinesis, this elongated peroxisome is split in two and both mother and daughter cells obtain part of this peroxisome. The actin-myosin ring (AMR) can be used as a marker for cytokinesis as it disappears just prior (1–2 min) to the deposition of the primary septum [61] that separates the mother cell cytoplasm from that of the daughter cell.

We analysed cells co-expressing the AMR marker Myo1-GFP and the peroxisomal membrane marker Pex11-mRuby2 in *vps1Δdnm1Δ* cells. The peroxisomal structure is divided within 1–2 min after the disappearance of the AMR (Appendix A). Presumably, the deposition of the primary septum cleaves the peroxisome as has been suggested for the nucleus [62]. This mechanism predicts that all *vps1Δdnm1Δ* cells would contain a single peroxisome; however, this is not the case (Appendix A). Time-lapse microscopy revealed that the elongated peroxisome is mobile (Appendix A), especially the end penetrating into the bud, and sometimes this end flips back into the mother cell (Appendix A). If this occurs around the time of cytokinesis, this leads to a segregation and fission defect (Appendix A). After approximately 280 min, multiple new peroxisomes appear in the cell that was devoid of peroxisomes. Previous observations in *vps1Δdnm1Δ* cells have shown that newly formed peroxisomes grow and segregate between mother and daughter cells, thereby reducing in number per cell with every cell division until a single peroxisome per cell is reached [45]. In line with these observations, a culture of dividing *vps1Δdnm1Δ* cells inherently shows a weak peroxisome segregation defect, with most cells containing a single enlarged peroxisome but a low percentage of cells lacking peroxisomes or containing multiple small peroxisomes that have formed de novo (Appendix A). In wild-type cells, no cells lacking peroxisomes are observed (Appendix A) [60].

### 3.2. Identification of Genes Affecting Peroxisome Dynamics Using a High-Content Microscopy Setup

Since *vps1Δdnm1Δ* cells inherently show a weak peroxisome segregation defect and mostly contain a single peroxisome, this strain provides a very sensitive phenotypic background to spot even minor effects on peroxisome dynamics caused by additional mutations in other genes. Hence, it is expected that a weak defect in peroxisome segregation will result in a population of cells firstly characterised by mispositioning of the single peroxisome in the budding cells, combined with an increase in cells with no peroxisomes and in cells with multiple de novo-formed small peroxisomes. We generated a tailor-made genome-wide yeast mutant collection in this genetic background through automated mating and a sporulation approaches platform [53,54] (for details on the construction of the mutant collection, see also the Materials and Methods section). The resulting library consists of strains containing mutations in three genes (*vps1Δdnm1Δxxx∆*) and expressing the fluorescent protein mNeonGreen, containing peroxisomal targeting signal type 1 and mNG-PTS1, along with cytosolic mCherry. These mutant strains were grown to log phase, imaged, and visually inspected to identify divergence from the *vps1Δdnm1Δ* phenotype using three different criteria: (i) mNG-PTS1 marker import, (ii) positioning of the peroxisome in the cell, and (iii) the number of peroxisomes.

Our initial analysis revealed 132 mutants that were aberrant in one or more of the criteria mentioned above. Subsequently, these mutants were classified into three major phenotypic classes (Figure 1B). Sixteen mutants (class 1) mis-localised mNG-PTS1 to the cytosol. Out of these 16, 14 are well-established peroxisome biogenesis (*PEX* gene) mutants that fail to import matrix proteins containing PTS1 (Appendix A). Unlike all other matrix protein import mutants, partial mNG-PTS1 localisation to peroxisomes was observed in a few *vps1Δdnm1Δpex17Δ* cells (Appendix A). The other two hits, *YJL211C* and *YGL152C*, are dubious open reading frames that partially overlap with *PEX2* and *PEX14*, respectively. This further corroborated the assumption that the annotated genes are not genuine hits but rather represent known peroxisomal protein deletions [63]. The cell population of the five class 2 mutants displayed a variable peroxisome number (Appendix A). Moreover, in these mutants, many cells mispositioned peroxisomes exclusively to either the mother cell or the daughter cell, away from their typical position at the bud neck. These phenotypes reflect typical inheritance mutants and, as expected, *INP1* and *INP2* were part of the class 2 mutants. Class 3 mutants displayed cells with multiple small peroxisomes. Many of the genes were associated with chromosomal maintenance or chromosomal segregation. Hence, there was a possibility that the increase in peroxisome number is a consequence of mis-segregation of the major peroxisomal DRP-encoding gene, *VPS1*, at the meiosis (sporulation) step during mutant library construction, resulting in aneuploidy of this gene. Therefore, the class 2 and 3 hits were tested for the presence of Vps1 by immunoblotting (Appendix A). The mutants that did not express Vps1 are included in the final hit list (Appendix A).

### 3.3. The SPOC Kinase Kin4 Is Required for Peroxisome Transport into the Bud

To identify the factors involved in peroxisome inheritance, we focused on class 2 mutants. In addition to *INP1* and *INP2*, the class 2 mutants also included *PEX25*, *KIN4*, and *MDL2*. The *vps1Δdnm1Δpex25Δ* cell population showed a mixed phenotype, with cells displaying a defect in peroxisome segregation and others in peroxisomal matrix protein import. Moreover, the peroxisome positioning in *vps1Δdnm1Δpex25Δ* cells is not exclusive to either the mother or the bud (Appendix A), as observed in *vps1Δdnm1Δinp2Δ* and *vps1Δdnm1Δinp1Δ* cells, respectively (Figure 2A). This phenotype was also observed when *PEX25* was first identified and is in line with a function in peroxisome multiplication and formation, as previously reported [44,60,64]. Mdl2 is an ABC half-transporter of the mitochondrial inner membrane [65] and was not further considered in this study. To test the reproducibility of the mutant phenotypes, the *vps1Δdnm1Δ* strains lacking either *INP1* or *INP2* or *KIN4* were logarithmically grown and analysed again by epifluorescence microscopy. In this case, peroxisome number distribution in *vps1Δdnm1Δkin4Δ* cells was comparable to the *vps1Δdnm1Δinp1Δ* cells and *vps1Δdnm1Δinp2Δ* cells but was different from the control, *vps1Δdnm1Δ* cells (Figure 2A,B). Further analysis revealed that in many *vps1Δdnm1Δkin4Δ* cells, peroxisomes are positioned exclusively in the mother cell, as observed in *vps1Δdnm1Δinp2Δ* cells (Figure 2A,D). Time-lapse imaging analysis of *vps1Δdnm1Δkin4Δ* cells showed a clear defect in peroxisome transport to the bud, and the cells devoid of peroxisomes eventually formed multiple small peroxisomes de novo after 3–4 h (Appendix A), which is very similar to what is observed ©n *vps1Δdnm1Δinp2Δ* cells (Appendix A). This is in contrast to *vps1Δdnm1Δ* cells, where mis-segregation is a very rare event, and hence, the de novo synthesis of peroxisomes is not triggered (Appendix A). We reasoned that the lack of peroxisomes in daughter *vps1Δdnm1Δkin4Δ* cells could be a consequence of either excessive retention by Inp1 or a defect in Inp2-dependent forward transport to the bud. To resolve this, the *INP1* and *INP2* genes were knocked out independently in *vps1Δdnm1Δkin4Δ* cells, and these quadruple mutant strains were analysed by microscopy. In line with previous observations that Inp1 is the main peroxisome retention factor, in the *vps1Δdnm1Δinp1Δ* strain, ~74% of mother cells were devoid of peroxisomes and peroxisomes ended up in the bud. Interestingly, in the *vps1Δdnm1Δkin4Δinp1Δ* strain, in only ~12% of cells, peroxisomes entered the bud. In contrast, there was no major difference in the distribution of peroxisomes in *vps1Δdnm1Δkin4Δinp2Δ* cells compared to *vps1Δdnm1Δinp2Δ* cells (Figure 2D). The above results suggest that Kin4 is involved in the transport of peroxisomes to the bud rather than a negative regulator of peroxisome retention in the mother.

### 3.4. Frk1 Is a Functional Paralog of Kin4

Kin4 function has not been implicated in any other organelle segregation, apart from the nucleus. Therefore, we sought to study *KIN4* deletion in a wild-type background instead of in a DRP-deficient mutant that inherently shows a weak peroxisome segregation defect. Analysis of *kin4Δ* cells showed that there was a considerable defect in peroxisome inheritance, especially in small-budded cells, but this was not as severe as in *inp2Δ* cells, where in most budding cells, the buds are devoid of peroxisomes (Figure 3A,B). This may explain why Kin4 was not identified before as a regulator of peroxisome dynamics. In a Saccharomyces Genome Database (SGD) search, it was found Kin4 has a paralogue, Frk1. Kin4 and Frk1 are members of the serine/threonine kinase family and share 43.6% identity and 57.7% similarity at the amino acid sequence level. Kin4 is positively regulated by the kinase Elm1 through phosphorylation of a threonine residue (T209) in the kinase activation loop [23,24]. This kinase activation loop motif is also strictly conserved in Frk1 (Figure 3C). Taken together, it was compelling to analyse *frk1Δ* and *kin4Δfrk1Δ* cells. Although *frk1Δ* cells did not show any noticeable defect in peroxisome segregation, the deletion of *KIN4* in *frk1Δ* cells resulted in a strong defect in peroxisome inheritance resembling that of *inp2Δ* cells (Figure 3A,B).

Accurate spindle alignment along the cell polarity axis is a prerequisite for faithful nuclear inheritance during mitosis. To avoid premature nuclear segregation in the mother due to a misaligned spindle, the spindle position checkpoint (SPOC) is activated. Kin4 is mainly localised to the mother cell and plays an established role in the SPOC, where it keeps the GTPase Tem1, the activator of the mitotic exit network (MEN), inactive. Kin4 achieves this through phosphorylation of Bfa1, a part of the bipartite GTPase-activating protein (GAP) complex Bfa1/Bub2 for Tem1 GTPase. Kin4 itself needs to be phosphorylated by Elm1 at T209 for its kinase activity in the SPOC (Figure 3D). Upon entry of an SPB into the bud, Bfa1/Bub2 activation through Kin4 is lost, and Tem1-GTPase can induce mitotic exit. Overexpression of *KIN4* is toxic to the cells since Kin4 is a negative regulator of the MEN and therefore blocks cellular exit from mitosis. Deletion of either *BFA1* or *ELM1* suppresses the lethal effect of *KIN4* overexpression [23,24]. On the other hand, Frk1 has not been previously implicated in the SPOC process. Hence, we tested whether overexpression of *FRK1* is toxic to cells and furthermore whether this is dependent upon Bfa1. *FRK1* was expressed in wild-type, *bfa1Δ*, and *elm1Δ* cells under the control of the strong inducible *GAL1* promoter. The transformants were grown in a raffinose-containing minimal medium before shifting to a galactose medium that induces expression. Indeed, wild-type cells expressing *FRK1* showed very little growth in the galactose medium and were perfectly fine in the glucose medium. Interestingly, *bfa1Δ* and *elm1Δ* cells expressing *FRK1* did not show such growth defects (Figure 3E,F). These results indicate that overexpressed Frk1 can activate Bfa1 similar to Kin4 and block the mitotic exit, thereby suggesting that both kinases are indeed paralogues and have residual overlapping activities.

The vacuole is another organelle in which transport to the emerging bud is solely dependent on Myo2 and the actin cytoskeleton. Therefore, we tested whether Kin4 and Frk1 are also required for vacuole inheritance. The vacuoles were visualised by staining the logarithmically growing cells with FM4-64 dye, followed by imaging with an epifluorescence microscope. In wild-type cells, clear vacuole staining was observed in both the mother and the bud (Figure 4A). Moreover, in small-budded cells, a segregation structure extending from the mother to the bud was also observed. Interestingly, these structures were mostly missing in *kin4Δfrk1Δ* cells, and consequently, many cells showed very faint or no FM4-64 staining in their bud; in some cases, the whole cells completely lacked FM4-64 staining (Figure 4B). These results suggest that Kin4 and Frk1 function not only in peroxisome, but also in vacuole inheritance. The homology with Kin4 at the amino acid sequence and at the functional role strongly suggests that Frk1 is a paralog of Kin4 at both the protein level and the functional level. Hence, the mechanistic insights gained from Kin4 and Frk1 functions in peroxisome inheritance are likely to also be applicable to the vacuole. In this study, we mainly studied peroxisomes as a proxy to decipher the role of Kin4 and Frk1 in organelle transport.

### 3.5. The Kinase Activity of Kin4 Is Required for Its Function in Peroxisome Segregation

As mentioned above, Elm1-dependent phosphorylation at T209 is important for Kin4 kinase activity [23] and its function in the SPOC [23,24]. Interestingly, the SPOC occurs at early anaphase, but T209 residue phosphorylation is observed throughout the cell cycle [23]. Therefore, it was intriguing to check whether Kin4 kinase activity is required to maintain the peroxisome positioning at the bud neck, as well as peroxisome segregation in *vps1Δdnm1Δ* cells subsequently. To test this, wild-type Kin4-GFP and the T209A mutant were expressed in *vps1Δdnm1Δkin4Δfrk1Δ* cells and were further analysed by epifluorescence microscopy. As expected, peroxisomes were mostly positioned in *vps1Δdnm1Δkin4Δfrk1Δ* cells in the mother cell, and as a consequence, 35.4% had multiple small peroxisomes (Figure 5A,B). The introduction of wild-type Kin4 alleviated the peroxisome inheritance defect by restoring the peroxisome position at the bud neck (Figure 5A,C), and this was reflected in the reduced number of cells (7.5%) with multiple small peroxisomes. In contrast, the expression of Kin4 carrying the T209A mutation failed to restore the peroxisome positioning at the bud neck, and hence, many cells still failed to inherit peroxisomes, successfully leading to 31.5% of cells with multiple small peroxisomes (Figure 5A,C). These results indicate that Kin4 kinase activity is required for its function in peroxisome inheritance.

### 3.6. kin4Δfrk1Δ Cells That Fail to Inherit Peroxisomes Form Them De Novo

In previous studies, it has been shown that cells lacking peroxisomes due to defects in inheritance can form them de novo [43]. Since peroxisome inheritance is affected in *kin4Δfrk1Δ* cells, the de novo formation of peroxisomes was tested in wild-type and *kin4Δfrk1Δ* cells. To test peroxisome de novo formation, constitutive mKate2-PTS1 and galactose-inducible mNG-PTS1 markers were expressed in wild-type and *kin4Δfrk1Δ* cells. In cells, mNG-PTS1 expression was induced by growing cells into a galactose-containing medium for 2.5 h and then shifted the cells to a glucose medium for 2 h to shut down the expression. Then, the cells were seeded under an agarose pad in mini-dishes and grown for an additional 6–8 h at 30 °C to allow colony formation. Subsequently, the cells were imaged using epifluorescence microscopy. The peroxisomes in wild-type colony cells were labelled with both mKate2 and mNG (Appendix A). This shows that the pre-existing pool of mNG-PTS1 present in the originally seeded cell passed on to progeny cells, indicating proper peroxisomes’ fission and segregation during cell division, and this is in line with previous observations [43,44]. In contrast, in the *kin4Δfrk1Δ* colony, this pre-existing pool of mNG-PTS1 was not properly passed on to progeny cells showing a failure in peroxisome partitioning. Most of the cells that did not inherit pre-existing peroxisomes formed peroxisomes de novo (mKate2-PTS1-labelled only) (Appendix A). A similar observation was made for *inp2Δ* cells that failed to inherit peroxisomes [43].

### 3.7. Kin4 Function in Peroxisome Inheritance Is Independent of Its Function in SPOC

Besides its role in the SPOC, Kin4 has not been implicated in the dynamics of other membrane-bound organelles. During activation of the SPOC pathway, Elm1 and Rts1 act upstream to activate Kin4, whereas Bfa1 and Bub2 act downstream (Figure 3D) [18]. In the genome-wide screen, no viable cells were observed when *ELM1* deletion was mated into *vps1Δdnm1Δ* cells. On the other hand, *vps1Δdnm1Δ* cells lacking either BFA1 or BUB2 or RTS1 were obtained in the screen. Interestingly, none of the above three genes were identified as affecting peroxisome dynamics. The triple mutants from the screen were subsequently regrown and analysed, and none of the triple mutants resembled the defect in either peroxisome number distribution or in peroxisome positioning, as observed in *vps1Δdnm1Δkin4Δ* cells (Figure 6A,B). In addition, *bfa1Δ* and *bub2Δ* cells did not show a peroxisome segregation defect (Appendix A). We conclude that the effect of Kin4 on peroxisome segregation is independent of its role in the SPOC.

Both Frk1 and Kin4 play a role in the SPOC and in peroxisome inheritance. As such, we asked if both SPOC activation and peroxisome inheritance affect each other. Therefore, GFP-Tub1 was expressed in wild-type, *kin4Δfrk1Δ*, and *kar9Δ* cells, and the orientation of the mitotic spindle was analysed. Kar9 is a spindle orientation factor that connects the astral microtubules with Myo2 and thus plays a crucial role in SPB alignment [66,67]. Wild-type and *kin4Δfrk1Δ* cells showed similar SPB alignment unlike *kar9Δ* cells, where in many cells, the mitotic spindle was misaligned. Moreover, *kar9Δ* cells are not affected in peroxisome inheritance (Figure 6C). The SPOC is activated only when the mitotic spindle is not aligned parallel to the cell polarity axis during early anaphase. We conclude that the peroxisome inheritance defect observed in *kin4Δfrk1Δ* cells is not a consequence of spindle misalignment and subsequent SPOC activation. In addition, overexpression of *KIN4* in *inp2Δ* is lethal as observed in wild-type cells, indicating that peroxisome inheritance is not required for SPOC activation (Figure 6D). Since neither *vps1Δdnm1Δbfa1Δ* nor *vps1Δdnm1Δbub2Δ* cells fail to transport peroxisomes to the bud, it supports that Kin4 and Frk1 do not signal through the SPOC pathway to mediate peroxisome inheritance. Moreover, the transport of peroxisomes is initiated in the G1 phase [4], which is much earlier in the cell cycle than SPOC signalling. All the above confirm that the function of Kin4 in peroxisome transport is independent of its role in the SPOC. We also conclude that SPOC activation and peroxisome inheritance do not affect each other.

### 3.8. Kin4 and Frk1 Are Required to Maintain Inp2 Steady-State Levels

The inheritance defect for peroxisomes in *kin4Δfrk1Δ* cells resembles the defect observed in mutants lacking Inp2. Hence, Inp2 protein levels in *kin4Δfrk1Δ* cells were compared to those in wild-type cells. We tagged Inp2 at its carboxy terminus as previous experiments have shown that this does not interfere with the Inp2 function [33]. To test protein levels, Protein A-tagged Inp2 (Inp2-ProtA) was expressed under its endogenous promoter in wild-type and *kin4Δfrk1Δ* cells. Interestingly, Western blot analysis revealed that the Inp2 protein levels were significantly reduced in *kin4Δfrk1Δ* compared to wild-type cells (Figure 7A,B). In contrast, Myo2-3 × HA levels were comparable in wild-type and *kin4Δfrk1Δ* cells (Figure 7C). This suggests that the inheritance defect is most likely a direct consequence of lowered Myo2 receptor protein levels.

If this was indeed the case, then Kin4 overexpression would lead to an increase in Inp2 protein levels, and as a consequence, would cause reduced peroxisome retention in the mother cell. To test this, *BFA1*-deficient cells were genomically modified to introduce the *GAL1* promoter upstream to *KIN4*. Western blot analysis revealed that iInp2-ProtA levels in *bfa1Δ-GAL-KIN4* were indeed increased upon galactose-induced Kin4 overexpression (Figure 7D). To study the peroxisome positioning and Inp2 localisation in this strain, cells were transformed with peroxisomal matrix protein markers, HcRed-PTS1 and Inp2-GFP, under endogenous promoter plasmids. Interestingly, in cells with *KIN4* overexpression, 14.3% of cells fail to retain peroxisomes in the mother, whereas without Kin4 overexpression, only 1 out of 114 cells did not retain peroxisome (Figure 7E). Furthermore, before galactose induction, Inp2-GFP was observed in 56.3% of cells and after galactose induction, this increased to 90.8% of cells. Further analysis revealed that more cells showed Inp2-GFP localisation at the bud neck with Kin4-overexpression than without (Figure 7F).

### 3.9. Kin4 and Frk1 Are Required to Maintain Elevated Inp2 Protein Levels in the Mother Cell

The class V Myo2 receptors on organelles harbour a Myo2 interacting site (MIS). In an earlier report, the crystal structure of the Inp2-MIS motif-containing peptide in complex with Myo2–cargo binding domain (Myo2-CBD) was determined, and mutations in the Inp2-MIS were demonstrated to reduce interaction with Myo2-CBD in vitro [68]. To validate these results in vivo, the wild-type and MIS lacking Inp2 proteins were expressed in *inp2Δ* cells and were analysed by epifluorescence microscopy. As expected, wild-type Inp2 restored peroxisome inheritance in *inp2Δ* cells comparable to wild-type cells. In contrast, the expression of Inp2-ΔMIS failed to do so (Figure 8A). Interestingly, Inp2-GFP was still localised to peroxisomes (Figure 8A). These results supported the function of the previously identified Inp2-MIS in vivo. The defect in receptor recognition by Myo2 not only led to failure in organelle transport to the emerging bud, but also elevated protein levels of Myo2 receptors (Inp2 and Vac17) [41,69]. This is indeed true in the case of the Inp2-ΔMIS mutant as well, as elevated protein amounts were detected in Western blot analysis for the Inp2-ΔMIS mutant compared to the wild-type version (Figure 8B,C). As established before, Kin4 is mainly localised to the mother cell; hence, it was compelling to analyse whether the elevated protein levels of the Inp2-ΔMIS mutant are affected by additional deletion of Kin4 and Frk1. Interestingly, Inp2-ΔMIS mutant protein levels were strongly reduced in the absence of both Kin4 and Frk1 (Figure 8D). This result shows that Kin4 and Frk1 are required to maintain elevated Inp2 levels in the mother cell.

## 4. Discussion

In eukaryotes, faithful segregation of organelles is essential to ensure their growth and survival under various metabolic stress conditions. Hence, several mechanisms have evolved to monitor proper organelle distribution during cell growth and division. In budding yeast, retention of organelles in the mother cell and transport to the emerging bud occur concomitantly. For peroxisomes, Inp1 and Inp2 are required for successful retention in the mother cell and for transport to the bud, respectively. Peroxisome segregation and fission are coupled with each other, where the fission process is mainly dependent on DRP Vps1 and, to a lesser extent, on Dnm1. Therefore, *vps1Δdnm1Δ* cells harbour mostly one elongated peroxisome that is retained with one end in the mother cell, with the other end being pulled into the bud. Upon cytokinesis, this single peroxisome is split in two. However, a small percentage (approximately 5–10%) of *vps1Δdnm1Δ* cells fail to position their peroxisome at the bud neck during cytokinesis, which leads to a segregation defect in those cells. At the cell population level, there is a weak inheritance defect.

This sensitised genetic background was exploited to perform a genome-wide high-throughput microscopy screen to identify molecular players that have not been implicated previously in peroxisome maintenance. The screen analysis revealed several mutants, along with *INP1* and *INP2*, that affected either peroxisome inheritance or abundance.

Among novel mutants, *KIN4* showed a reproducible strong phenotype and hence was studied further to understand its role in peroxisome maintenance. In *vps1Δdnm1Δkin4Δ* cells, peroxisomes frequently fail to be delivered to the daughter cell. This phenotype is due to a defect in forward transport to the bud rather than excessive retention in the mother. A *KIN4* deletion in cells that are proficient in DRP-mediated peroxisome fission is also affected in peroxisome segregation, with a clear delay of transport of peroxisomes to buds, i.e., small buds are frequently found to lack peroxisomes. However, large-budded cells frequently do obtain peroxisomes. This may explain why Kin4 had not been identified in previous genetic screens and has not been implicated in organelle transport. However, we showed that Frk1, a Kin4 paralog, also contributes to the peroxisome inheritance, as in *kin4Δfrk1Δ* cells, in which peroxisome transport to the bud is strongly affected and resembles that of *inp2Δ* cells. In addition to peroxisome transport, we found that Kin4 and Frk1 are also required for vacuole transport.

Kin4 is a well-characterised SPOC kinase and is required for mitotic spindle alignment maintenance [21,70]. However, the functional roles of Kin4 in the SPOC and in peroxisome inheritance are distinct as the defect in peroxisome transport is not a consequence of the Kin4 requirement in the SPOC and peroxisome transport is dispensable for Kin4 activity in the SPOC (Figure 9A). Overexpression of Kin4 is lethal to the cells and we found that overexpression of Frk1 also inhibits growth. In both cases, the toxicity caused due to overexpression can be rescued by deletion of either *BFA1* or *ELM1*. This demonstrated that Frk1 is, in addition to Kin4, a potential SPOC kinase. Moreover, Frk1 shares significant identity and similarity at the amino acid sequence level with Kin4; therefore, we concluded that Frk1 is a Kin4 paralog. From a future perspective, it is intriguing to test if Frk1 plays a role in the SPOC and thus eventually regulates MEN. This could be tested by analysing whether Frk1 is a direct substrate of the kinase Elm1 and whether Frk1 can phosphorylate Bfa1 in vivo.

Next, we showed that Kin4 kinase activity is required for peroxisome inheritance. Kin4 and Frk1 contribute to peroxisome inheritance by maintaining Inp2 protein steady-state levels (Figure 9B). In line with this, Kin4 overexpression led to increased Inp2 levels in the cells. We validated the in vivo function for the previously identified Inp2-MIS. In addition, we also demonstrated that the elevated levels of Inp2ΔMIS mutants are strongly reduced in the absence of Kin4 and Frk1, further corroborating the role of these two kinases in regulating Inp2 protein levels. In large-budded cells, Myo2 moves to the bud neck along actin cables and stays there throughout the cytokinesis stage as new buds emerge in the vicinity. In the case of Vac17-breakdown deficient mutants, like in *dma1Δdma2Δ* cells, Vac17 also accumulates to the bud neck, leading to vacuole mispositioning. Interestingly, peroxisomes are also mispositioned at the bud neck in *dma1Δdma2Δ* cells [6], indicating similar pathways and components shared for peroxisome and vacuole transport. Dma1/2 activity is positively regulated by p21 kinases (PAKs), Cla4, and Ste20. Therefore, in *cla4*/*ste20*-deficient cells Vac17 also localises to the bud neck [7]. In this study, we report similar observations for Inp2-GFP localisation in large-budded cells upon Kin4 overexpression. Moreover, Cla4 and Kin4 act antagonistically in the SPOC. Therefore, it is very tempting to speculate that Kin4 regulates Inp2 protein levels antagonistically with the Cla4/Ste20 and Dma1/2 pathway. However, this needs to be confirmed by further experiments.

## Figures and Tables

**Figure 1 biomolecules-13-01098-f001:**
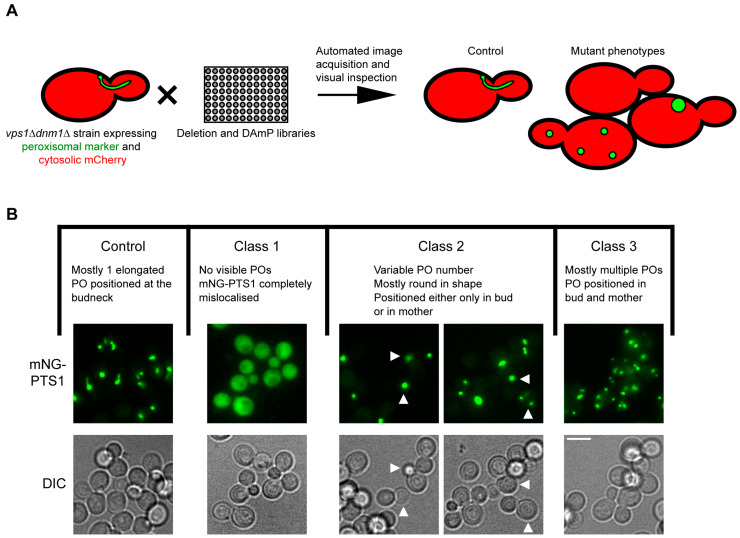
A genome-wide genetic screen to identify novel factors regulating peroxisome dynamics. (**A**) Schematic depiction of the genome-wide screen layout. The *vps1Δdnm1Δ* strain (mating type α) expressing genomically integrated peroxisomal mNG-PTS1 and cytosolic mCherry was crossed with the library of deletion mutants and the library of hypomorphic alleles of essential genes (DAmP). Both mutant libraries are mating type **a**. Sporulation was induced, and haploids expressing both fluorescent marker proteins and containing *VPS1* and *DNM1* gene deletion, as well as deletion of the library mutant, were selected. Automated imaging of the arrayed collection was performed on the growing cell cultures, and the obtained images were analysed manually. (**B**) Summary of the imaging analysis. The three observed main phenotypes are indicated by description, schematic depictions, and exemplified by a representative image. ‘‘Control’’ represents the phenotype obtained with most triple mutants and is identical to the phenotype observed in the control *vps1Δdnm1Δ* cells. The ‘‘class 1, 2, and 3′’ mutants’ phenotypes were considered hits. Arrowheads in the class 2 mutant panel indicate either bud (left) or mother (right). Pos: Peroxisomes. For the complete list of hits, see Appendix A. Cells from exponentially growing cultures were used for all epifluorescence microscopy experiments. Scale bar, 5 µm.

**Figure 2 biomolecules-13-01098-f002:**
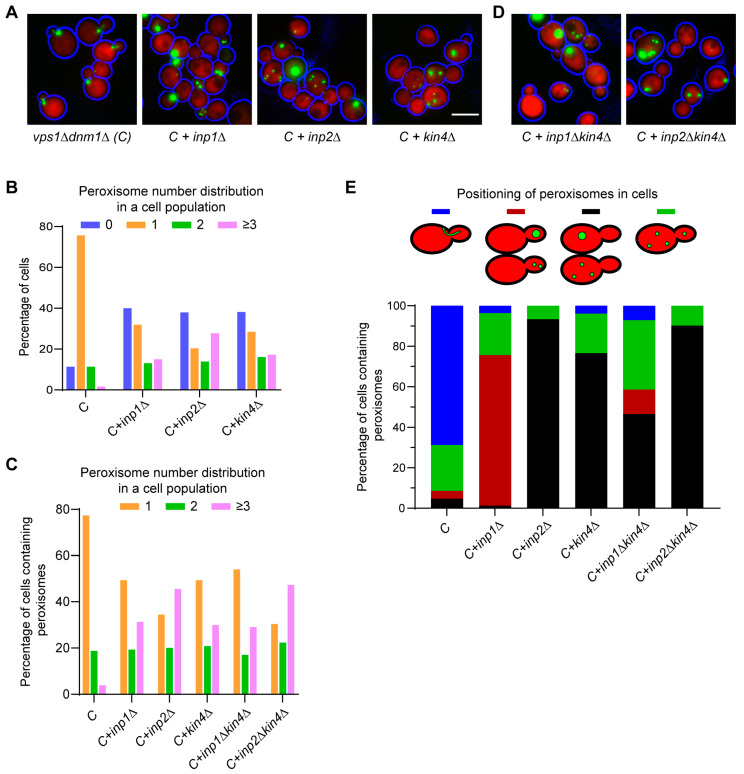
SPOC kinase Kin4 is required for peroxisome transport to the bud. (**A**) The *vps1Δdnm1Δ* cells lacking *INP1*, *INP2,* or *KIN4* and expressing mNG-PTS1 and cytosolic mCherry were analysed using an epifluorescence microscope. (**C**) Control phenotype of the *vps1Δdnm1Δ* cells. (**B**) Quantification of peroxisome distribution in strains in (**A**). A minimum of 108 cells were analysed per strain. (**C**) Quantification of peroxisome distribution in cells (at least n = 77) containing mNG-PTS1 punctate patterns from strains in (**A**,**D**). (**D**) Representative epifluorescence images of *vps1Δdnm1Δinp1Δkin4Δ* and *vps1Δdnm1Δinp2Δkin4Δ* cells. (**E**) Quantitation of peroxisome positioning in the cells (at least n = 77) from strains in (**A**,**D**). Cells expressing mNG-PTS1 and cytosolic mCherry from exponentially growing cultures were used for all epifluorescence microscopy experiments. Bright-field images were collected in one plane and processed to highlight the cell circumference in blue. Scale bar, 5 µm.

**Figure 3 biomolecules-13-01098-f003:**
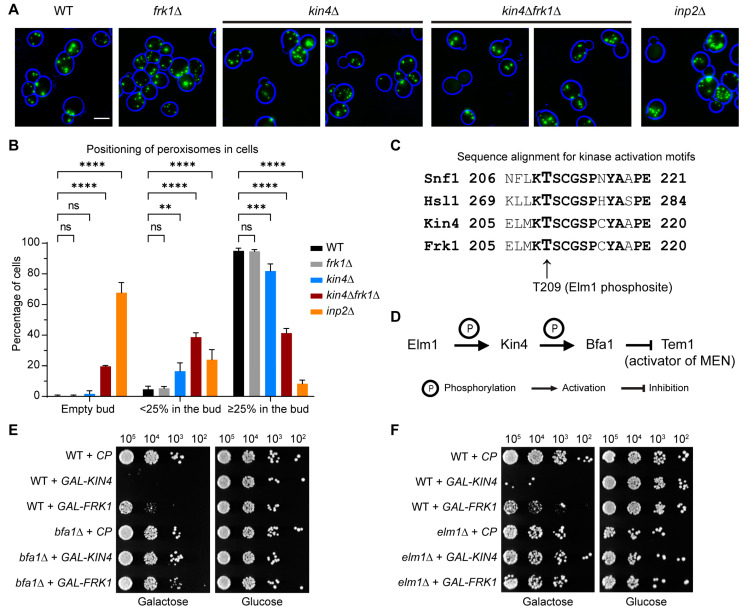
Frk1 is a functional paralog of Kin4. (**A**) Epifluorescence images were captured from the various exponentially growing *S. cerevisiae* mutant cells expressing mNG-PTS1. Representative images are shown as merged z-stacks. Bright-field images were collected in one plane and processed to highlight the cell circumference in blue. Scale bar, 5 µm. (**B**) Peroxisome quantitation of cells from the strains in (**A**). A minimum of 100 cells were analysed from 3 independent experiments. Statistical significance was determined using two-way ANOVA test. ns: not significant, ** *p* < 0.01, *** *p* < 0.001, **** *p* < 0.0001. Error bars indicate SD: standard deviation. (**C**) Sequence alignment for the activation loop motifs in Elm1 substrate kinases. Elm1 phosphorylates Kin4 at T209, which is strictly conserved in Frk1. (**D**) Schematic representation for SPOC pathways components and their regulation. Serial dilutions of wild-type, *bfa1Δ* (**E**), and *elm1Δ* (**F**) cells overexpressing either *KIN4* or *FRK1* controlled by the *GAL1* promoter were spotted on yeast minimal (YM) media plates containing either galactose/glucose as a sole carbon source. The plates were incubated for 3 days at 30 °C before imaging.

**Figure 4 biomolecules-13-01098-f004:**
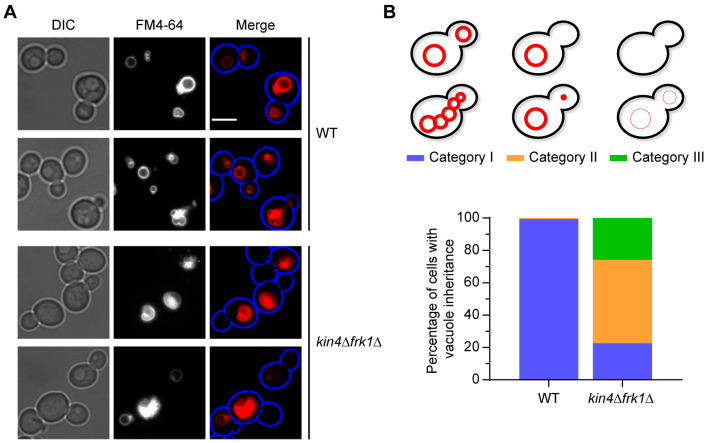
Vacuole inheritance is strongly affected in *kin4Δfrk1Δ* cells. The cells were grown to log phase and incubated with FM4-64 to stain the vacuoles, followed by imaging with epifluorescence microscopy. (**A**) Representative images for wild-type and *kin4Δfrk1Δ* cells are shown. Bright-field images were collected in one plane and processed to highlight the cell circumference in blue. Scale bar is 5 μm. (**B**) A minimum of 170 cells of each strain were scored to analyse the defect in vacuole inheritance. Categories I, II, and III represent the variable FM4-64 staining of the vacuoles in the cells.

**Figure 5 biomolecules-13-01098-f005:**
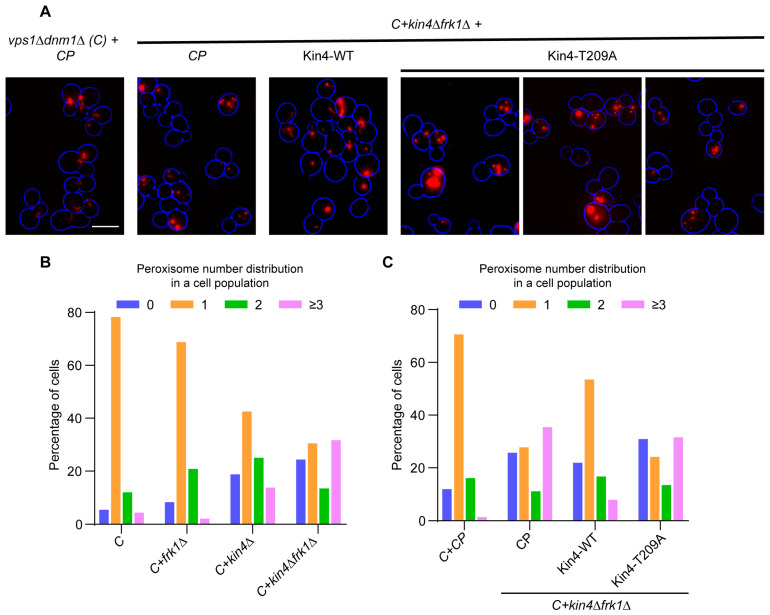
Kinase activity of Kin4 is required for its function in peroxisome segregation. Kin4 and Kin4-T209A were expressed under native promoter in *vps1Δdnm1Δkin4Δfrk1Δ* cells. (**A**) Representative epifluorescence images for exponentially growing cells are shown. Bright-field images were collected in one plane and processed to highlight the cell circumference in blue. (**C**) *vps1Δdnm1Δ*. Scale bar is 5 μm. (**B**) Quantitation of peroxisome distribution in *vps1Δdnm1Δ* cells lacking either *KIN4* or *FRK1* or both. (**C**) Quantitation of peroxisome distribution in *vps1Δdnm1Δkin4Δfrk1Δ* cells after Kin4 and Kin4-T209A reintroduction.

**Figure 6 biomolecules-13-01098-f006:**
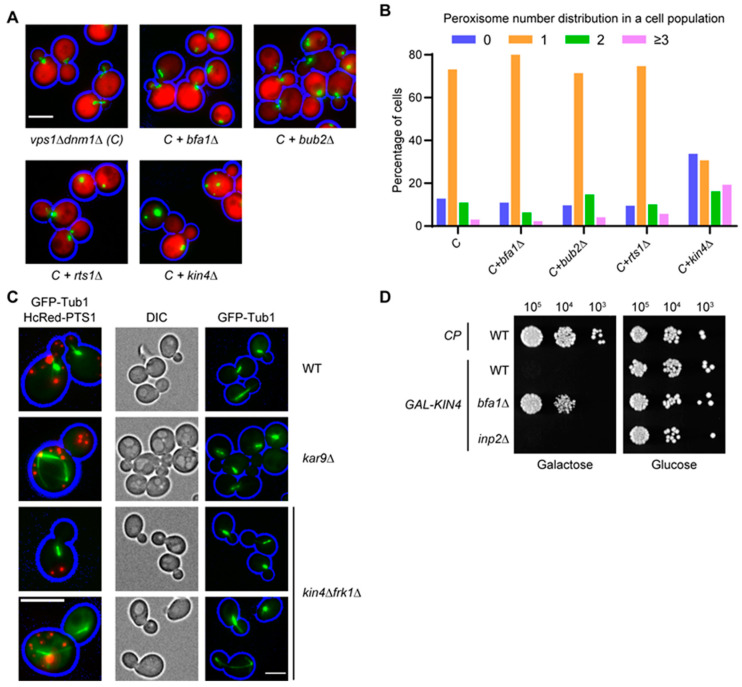
Kin4-dependent SPOC activation and inheritance of peroxisomes are independent processes. (**A**) Representative epifluorescence images of *vps1Δdnm1Δ* cells lacking *BFA1*, *BUB2,* or *RTS1* and expressing mNG-PTS1 and cytosolic mCherry. (**B**) Peroxisome distribution in strains in (**A**). A minimum of 158 cells were analysed per strain. Cells from exponentially growing cultures were used for epifluorescence microscopy experiments. C: *vps1Δdnm1Δ*. Scale bar, 5 µm. (**C**) Mitotic spindle alignment in *kin4Δfrk1Δ* cells is unaffected. Cells expressing GFP-Tub1 or/and mKate2-PTS1 were grown to log phase and were analysed by epifluorescence microscopy. (**D**) Peroxisome inheritance is not required for SPOC activation. Serial dilutions of wild-type, *bfa1Δ,* and *inp2Δ* cells overexpressing *KIN4* controlled by the *GAL1* promoter were spotted on yeast minimal (YM) media plates containing either galactose/glucose as a sole carbon source. The plates were incubated for 3 days at 30 °C before imaging. Epifluorescence images from exponentially grown cells expressing mNG-PTS1. Bright-field images were collected in one plane and processed to highlight the cell circumference in blue. Scale bar, 5 µm.

**Figure 7 biomolecules-13-01098-f007:**
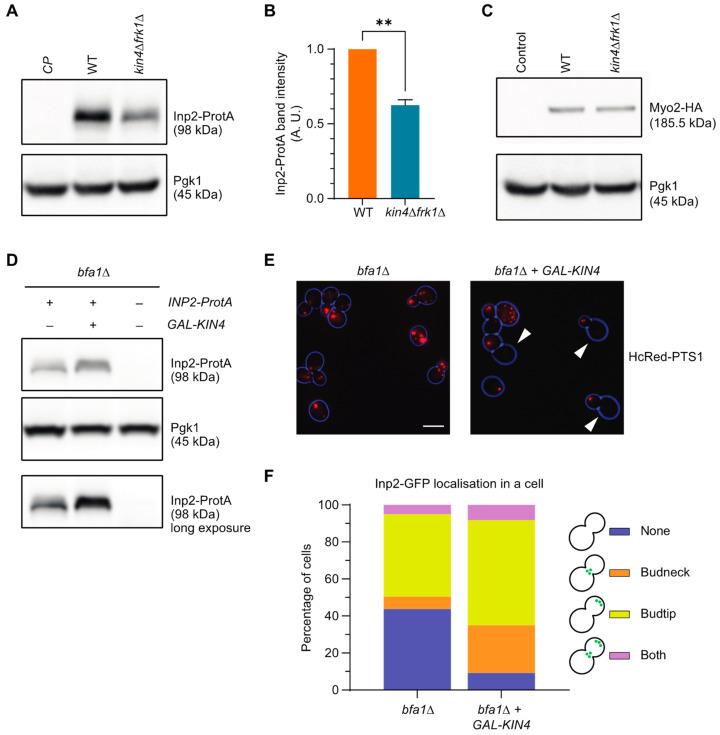
Kin4 and Frk1 are required to maintain Inp2 protein steady-state levels. (**A**) Inp2-ProtA was expressed in wild-type and *kin4Δfrk1Δ* cells. Cell lysate extracts were analysed by Western blot. (**B**) Intensity values for Inp2-ProtA bands were normalised against unsaturated Pgk1 bands and were plotted. Normalised ProtA signals in wild-type cells were set to 1 A. U. where A. U. is an arbitrary unit. (**C**) Myo2 was tagged with 3xHA in the genome. TCA extraction followed by Western blot analysis was performed on modified wild-type and *kin4Δfrk1Δ* strains. Error bars indicate SEM (standard error mean). N = 3. ** *p* < 0.01; two-tailed Student’s *t*-test. (**D**) Inp2-ProtA was expressed under native promoter in *bfa1Δ* and *bfa1Δ+GAL-KIN4* cells. Cell lysate extracts were analysed by Western blot. (**E**) Representative epifluorescence images of *bfa1Δ* and *bfa1Δ+GAL-KIN4* cells expressing HcRed-PTS1 peroxisomal marker. Cells from exponentially growing cultures were used for epifluorescence microscopy experiments. Bright-field images were collected in one plane and processed to highlight the cell circumference in blue. Arrowheads indicate mother cells lacking peroxisomal labelling. Scale bar, 5 µm. (**F**) Quantitation of Inp2-GFP localisation upon KIN4 overexpression after galactose induction in *bfa1Δ*. Inp2-GFP localisation was grouped into four categories based on Inp2-GFP accumulation in the cell. A minimum of 119 cells were analysed for each strain.

**Figure 8 biomolecules-13-01098-f008:**
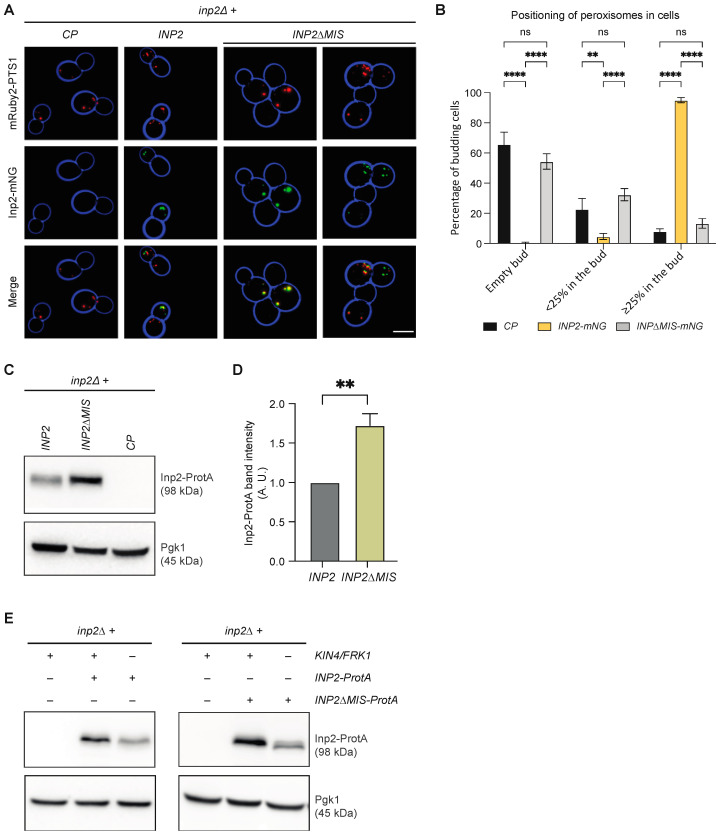
Kin4 and Frk1 are required to maintain elevated Inp2ΔMIS mutant protein levels. (**A**) Representative epifluorescence images of *inp2Δ* cells expressing c-terminally mNG-tagged Inp2 wild-type and ΔMIS mutants under endogenous promoter and mRuby2-PTS1 peroxisomal marker. Cells from exponentially growing cultures were used for epifluorescence microscopy experiments. Bright-field images were collected in one plane and processed to highlight the cell circumference in blue. Scale bar, 5 µm. (**B**) Quantitation of peroxisome positioning in the cells from strains in (**A**). A minimum of 100 cells were analysed from 3 independent experiments. Statistical significance was determined using two-way ANOVA test. ns: not significant, ** *p* < 0.0029, **** *p* < 0.0001. Error bars indicate SD: standard deviation. (**C**) C-terminally ProtA-tagged Inp2 wild-type and ΔMIS mutants were expressed in wild-type and *inp2Δ* cells. Cell lysate extracts were analysed by Western blot. (**D**) Intensity values for Inp2-ProtA bands were normalised against unsaturated Pgk1 bands and were plotted. Normalised ProtA signals in wild-type cells were set to 1 A. U. where A. U. is an arbitrary unit. Statistical significance was determined using two-way ANOVA test. ** *p* < 0.0085, (**E**) C-terminally ProtA-tagged Inp2 wild-type and ΔMIS mutants were expressed in wild-type, *inp2Δ*, and *kin4Δfrk1Δ* cells. Cell lysate extracts were analysed by Western blot.

**Figure 9 biomolecules-13-01098-f009:**
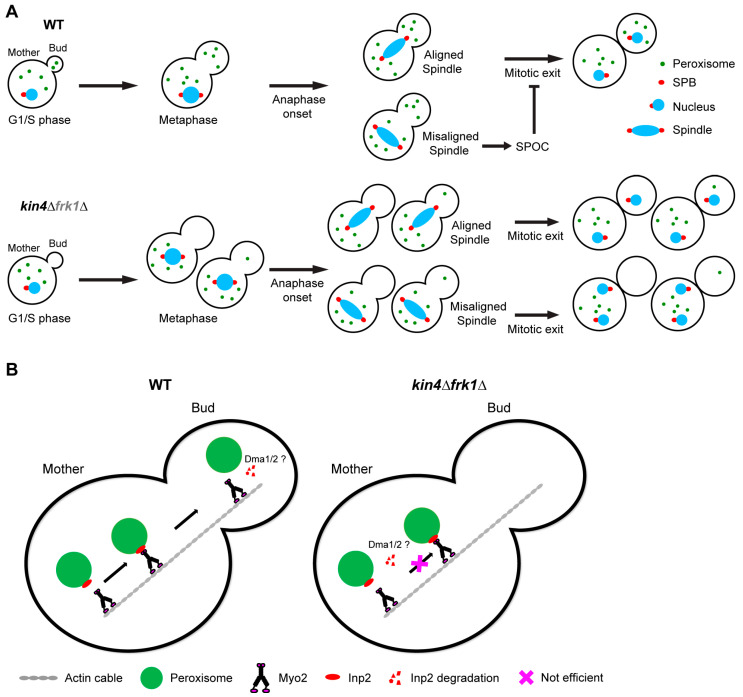
Schematic model showing the independent roles of Kin4 and Frk1 in peroxisome transport and SPOC. (**A**) Kin4 and its functional paralog, Frk1, are required for peroxisome transport to the emerging bud. In addition, Kin4 is a key component of SPOC and thus negatively regulates mitotic exit in case of a misaligned spindle. Peroxisome transport is not essential for the activation of SPOC and MEN and vice versa. Though Frk1 (in grey) is a potential SPOC kinase candidate like Kin4, its direct involvement in SPOC requires further investigation. (**B**) Kin4 and Frk1 contribute to peroxisome transport by maintaining steady state levels of the Myo2 receptor, Inp2, on peroxisomes, thus facilitating stable/efficient Inp2-Myo2 complex formation and active and successful peroxisome transport to the bud. Dma1/2 have been implicated in Inp2 degradation in the bud to terminate peroxisome transport, ensuring directional inheritance of peroxisomes. Here, we propose that, in the absence of Kin4 and Frk1, Dma1/2 recognises Inp2 while it is in the mother cell and leads to its premature degradation, inhibiting peroxisome movement to the bud. SPOC: spindle position checkpoint; MEN: mitotic exit network.

## Data Availability

All data are contained in the manuscript and Appendix A.

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
