# Peer review of "Spindle Position Checkpoint Kinase Kin4 Regulates Organelle Transport in Saccharomyces cerevisiae"

_biomolecules, 2023, doi:10.3390/biom13071098_

Round 1
Reviewer 1 Report
In this paper the authors present interesting new findings (summarized below) which are based on complicatd but well thought-out methodology (the SGA screen) and more up to date methods of yeast genetic engineering and fluorescence image analysis. The main result is the role of Kin4 in the regulation of the transport of peroxisomes to the daughter cell (bud), and then through a series of quite elegant experiments and genetic reasoning, the near complete independence of this role from from its other, previously known role in the position of the mitotic spindle and the SPOC (spindle orientation checkpoint). The functions described for Kin4 are in a certain sense, conditional, because there is more than one way of peroxisome inheritance and de novo synthesis in the cell division cycle. This made the investigation more difficult. The later part of the paper is well-written.
Here come my points of critique:
The journal Biomolecules is not a genetics journal and the paper should be understandable for readers with a general background in yeast cell biology. This would require a better explanation of how the SGA sceen works what it can find and will miss to find. The genetic network behind peroxisome inheritance and the entry into mitosis is fairly complicated. It would help to include a glossary of the more than 20 genes (three latter gene names) mentioned in the text and their known functions. Otherwise many non-specialst readers would be lost.
The screening procedure with the many selection markers and screening markers must be described in more detail. This is true in particular for the query strain. Why combine in the query strain a double deletion of dnm1 and vps1? What is the expected aoutcome: which not previously known genes coding for components of the transport system of peroxisomes on actin filamnents can be expected?
Genetic nomenclature is non-standard (use of the slash in the abbreviatd description of the strains). As mentioned before, most of the readers will be non-specialists for refined yeast genetics methods, therefore please more detail and use the standard way of strain descriptions.
Results as shown in the fluorescence pictures are excellent. More explanations for the general reader would be very welcome.
I like the way how the relationship between the two paralogs, Kin4 and Frk1, is clarified and how the relation and parallelism of vacuole and peroxisome inheritance is shown.
In the discussion part please draw a picture of the the importance of these findings for the regulation of the mitotic exit network: cells should separate only after the bud has received a share of the peroxisomes and only after the transport machinery has been inactivated by the ubiquitin proteasome system.
Please discuss those gene combinatins in this screen, which were lethal or very sick. They are presently not mentioned.
In summary my verdict is: Major revision, no new experiments are needed but a better description of the method and of the details of the screening procedure used.
Author Response
please find point by point response attached

Reviewer 2 Report
The manuscript authored by Dr. Ekal and colleagues is a comprehensive study aimed to gain a better understanding of the spatiotemporal regulation of actomyosin-based organelle transport in yeast, using peroxisome inheritance as a proxy for this purpose. By carrying out a genome-wide genetic screen in vps1Δ/dnm1Δ cells, which show a weak peroxisome segregation defect, the authors identified the spindle position checkpoint kinase Kin4 (and to a lesser extent also its paralog Frk1) as a regulator of peroxisome inheritance. In addition, the authors demonstrated that (i) the role of Kin4 in peroxisome transport can be functionally uncoupled from its mitotic surveillance mechanism, (ii) peroxisome segregation requires the kinase activity of Kin4, (iii) Kin4 and Frk1 regulate peroxisome transport by protecting the peroxisomal Myo2 receptor Inp2 from premature degradation in the mother cell, and (iv) kin4Δ/frk1Δ cells that fail to inherit peroxisomes form them de novo. Finally, the authors provide evidence that Kin4 and Frk1 also function in vacuole inheritance, thereby suggesting that the transport of peroxisomes and vacuoles along actin filaments shares a common regulatory mechanism. In summary, this study establishes Kin4 (and to a lesser extent also Frk1) as an important player in actomyosin-based organelle transport in S. cerevisiae. However, before acceptance, the authors should address the following minor comments:
1. Lines 121 and 277 – Rename the supplementary figures according to their appearance.
2. Lines 160 and 163 – Define SGA and DAmP (and explain these terms briefly to those unfamiliar with yeast genetics).
3. Line 334 – “(A) The vps1Δ/dnm1Δ 334 cells lacking INP1, INP2 and KIN4”. Replace “and” by “or”.
4. Can the authors hypothesize (e.g., in the discussion) on the putative mechanisms of how Kin4/Frk1 activity mediates Inp2 stability?
Author Response
Reviewer 2: Comments and Suggestions for Authors
The manuscript authored by Dr. Ekal and colleagues is a comprehensive study aimed to gain a better understanding of the spatiotemporal regulation of actomyosin-based organelle transport in yeast, using peroxisome inheritance as a proxy for this purpose. By carrying out a genome-wide genetic screen in vps1Δ/dnm1Δ cells, which show a weak peroxisome segregation defect, the authors identified the spindle position checkpoint kinase Kin4 (and to a lesser extent also its paralog Frk1) as a regulator of peroxisome inheritance. In addition, the authors demonstrated that (i) the role of Kin4 in peroxisome transport can be functionally uncoupled from its mitotic surveillance mechanism, (ii) peroxisome segregation requires the kinase activity of Kin4, (iii) Kin4 and Frk1 regulate peroxisome transport by protecting the peroxisomal Myo2 receptor Inp2 from premature degradation in the mother cell, and (iv) kin4Δ/frk1Δ cells that fail to inherit peroxisomes form them de novo. Finally, the authors provide evidence that Kin4 and Frk1 also function in vacuole inheritance, thereby suggesting that the transport of peroxisomes and vacuoles along actin filaments shares a common regulatory mechanism. In summary, this study establishes Kin4 (and to a lesser extent also Frk1) as an important player in actomyosin-based organelle transport in S. cerevisiae. However, before acceptance, the authors should address the following minor comments:
- Lines 121 and 277 – Rename the supplementary figures according to their appearance.
? Renaming has now been done.
- Lines 160 and 163 – Define SGA and DAmP (and explain these terms briefly to those unfamiliar with yeast genetics).
? We added more details with references.
- Line 334 – “(A) The vps1Δ/dnm1Δ 334 cells lacking INP1, INP2 and KIN4”. Replace “and” by “or”.
? We have replaced this.
- Can the authors hypothesize (e.g., in the discussion) on the putative mechanisms of how Kin4/Frk1 activity mediates Inp2 stability?
? We have now added a model (Fig. 9), which depicts our hypothesis without being too over-reaching.
Round 2
Reviewer 1 Report
I have carefully studied the revised version of the manuscript submitted by Ekal et al.
The authors have improved the manuscript very much and have added the parts which I suggested when viewing version 1. Therefore, I judge that the paper should be acceptable for publication now in its new form. The scheme shown in Fig. 9B is easy to understand and explains very well the authors‘ model for the role of Kin4 etc. in peroxisome transport to the bud. The manuscript contains very few errors: In Fig. 1 „varibale“ should be „variable“.